# Challenges in Imaging Analyses of Biomolecular Condensates in Cells Infected with Influenza A Virus

**DOI:** 10.3390/ijms242015253

**Published:** 2023-10-17

**Authors:** Temitope Akhigbe Etibor, Aidan O’Riain, Marta Alenquer, Christian Diwo, Sílvia Vale-Costa, Maria João Amorim

**Affiliations:** 1Cell Biology of Viral Infection Lab (CBV), Instituto Gulbenkian de Ciência (IGC), Fundação Calouste Gulbenkian, R. Quinta Grande, 6, 2780-156 Oeiras, Portugal; etibor@igc.gulbenkian.pt (T.A.E.); aoriain@igc.gulbenkian.pt (A.O.); malenquer@igc.gulbenkian.pt (M.A.); cdiwo@igc.gulbenkian.pt (C.D.); svcosta@igc.gulbenkian.pt (S.V.-C.); 2Cell Biology of Viral Infection Lab (CBV), Católica Biomedical Research Centre (CBR), Católica Medical School, Universidade Católica Portuguesa, Palma de Cima, 1649-023 Lisboa, Portugal

**Keywords:** biomolecular condensates, imaging, virology, influenza A virus

## Abstract

Biomolecular condensates are crucial compartments within cells, relying on their material properties for function. They form and persist through weak, transient interactions, often undetectable by classical biochemical approaches. Hence, microscopy-based techniques have been the most reliable methods to detail the molecular mechanisms controlling their formation, material properties, and alterations, including dissolution or phase transitions due to cellular manipulation and disease, and to search for novel therapeutic strategies targeting biomolecular condensates. However, technical challenges in microscopy-based analysis persist. This paper discusses imaging, data acquisition, and analytical methodologies’ advantages, challenges, and limitations in determining biophysical parameters explaining biomolecular condensate formation, dissolution, and phase transitions. In addition, we mention how machine learning is increasingly important for efficient image analysis, teaching programs what a condensate should resemble, aiding in the correlation and interpretation of information from diverse data sources. Influenza A virus forms liquid viral inclusions in the infected cell cytosol that serve as model biomolecular condensates for this study. Our previous work showcased the possibility of hardening these liquid inclusions, potentially leading to novel antiviral strategies. This was established using a framework involving live cell imaging to measure dynamics, internal rearrangement capacity, coalescence, and relaxation time. Additionally, we integrated thermodynamic characteristics by analysing fixed images through Z-projections. The aforementioned paper laid the foundation for this subsequent technical paper, which explores how different modalities in data acquisition and processing impact the robustness of results to detect *bona fide* phase transitions by measuring thermodynamic traits in fixed cells. Using solely this approach would greatly simplify screening pipelines. For this, we tested how single focal plane images, Z-projections, or volumetric analyses of images stained with antibodies or live tagged proteins altered the quantification of thermodynamic measurements. Customizing methodologies for different biomolecular condensates through advanced bioimaging significantly contributes to biological research and potential therapeutic advancements.

## 1. Introduction 

Imposing controlled phase transitions to biomolecular condensates holds great promise for the next generation therapies for cancer, neurodegeneration, and infections [1,2]. This potential stems from the role of biomolecular condensates as specialised compartments that rely on distinct material properties for activity [3,4,5,6,7,8]. For this, it is important to be able to perform quantitative and robust high-throughput screens to search for compounds and cellular factors that are able to alter the material properties or dissolve condensates [4,7,8]. Conventional biochemical tools lack the sensitivity to identify interacting partners in biomolecular condensates due to the weak interactome nature within these structurers [9,10,11] and many innovative methods being under development, but their validation is still complex [12,13,14,15,16]. To date, most of the studies investigating the biology, formation, material properties (and their alterations) and function of biomolecular condensates have employed microscopy-based approaches.

The increasing importance of bioimaging in the life sciences has underlined the pressing demand for technological advances capable of overcoming the limitations of image-based quantifications. Image analyses, regardless of being associated with high-throughput computer processing, involves image acquisition and powerful analyses that require custom workflows, including segmentation. This ensures the accurate identification of intracellular structures, encompassing classical membrane-bound organelles and biomolecular condensates, including those of viral origin found in infected cells, including viral inclusions, viral factories, virosomes, viroplasms, negri-bodies, and others [17,18,19,20]. Viral inclusions, such as those observed in influenza A virus (IAV) infection, are liquid condensates that accumulate progeny RNA [21,22,23,24,25,26,27], presumably to facilitate the assembly of its eight-partite genomic complex [8,28]. This liquid state raises a new hypothesis on how influenza epidemic and pandemic genome assembly takes shape. The study of the material properties of these structures involves using live cell imaging to assess the dynamics, viscosity, and internal molecular flexibility within viral inclusions. Both live and fixed images are utilised to measure thermodynamic parameters, determining whether the system is dissolving or concentrating more material within the condensates in response to alterations [1,29,30,31,32,33,34]. These measurements encompass various aspects, including area/volume, shape, material concentration within viral inclusions (Cdense), and in the surrounding media (Cdilute), as has been previously conducted [1,29,30,31,32,33,34].

However, analysing viral inclusions through imaging poses challenges due to their variable size (from 100 nm to 1µm diameter) and dynamic behaviours, complicating image acquisition and analysis quality. Some inclusions exhibit high dynamism, continuously fusing and dividing, while others remain relatively static, steadily receiving and sending small amounts of material. The choice of methodology, the type of microscope, time intervals between frame acquisition, achieved resolution level, and sample processing techniques may profoundly impact the quality and interpretation of acquired images. For example, confocal microscopy typically achieves a resolution of around 250 nm in the x and y directions, whereas super-resolution microscopy can enhance this to as fine as 50 nm [35]. Consequently, conventional confocal microscopy may bias acquisition towards larger viral inclusions.

Given the critical role of microscopy in studying biomolecular condensates and the high dependence on it, understanding how image acquisition and processing impact data and their interpretation is pivotal. Factors such as the use of live or fixed images and analysing a single focal plane versus an entire cell (as a Z-projection or 3D volume) remain unclear in their effect on results. Different analytical methodologies chosen for fundamentally the same sample can lead to distinct results and biological interpretations (Figure 1). For example, distorted oval shapes may be misinterpreted as rounded structures when Z-projections are employed (Figure 1, example 1). Additionally, different numbers and shapes may result from analysing a single focal plane compared to a Z-projection (Figure 1, example 2). In such an example, Z-projections may compile fluorescence to display a single large condensate, when in reality, it may be several smaller condensates that differ in space only by their Z plane, which are then stacked on top of each other as one. The interpretation can also be affected by the compounds used to recognise structures of interest, such as nanobodies, antibodies, or tagged proteins, each with distinct limitations and impacts on analyses and data relationships. Reports have dealt with issues surrounding the fixation rate of samples [36], and the consensus is that live cell imaging should be used as much as possible. However, it is widely known that for this, the structure of interest needs to be fused to a fluorescent protein, which may impose differences in the function, conformation, and interactome of the original protein, which may interfere with the ability to form biomolecular condensates. In addition, for viruses, tagging viral proteins often results in viral attenuation, which changes the concentration of proteins, a pivotal regulatory element in phase separation [37,38]. 

To address some of these concerns, we utilised the IAV inclusion model to elucidate potential technical challenges that may arise when studying biomolecular condensates via microscopy-based image analysis. We investigate whether the values calculated for various thermodynamic variables are influenced by the specific sample processing and image analyses methods employed to narrow this gap in knowledge. We describe the advantages, challenges, and limitations of using distinct methodologies in image-based processing and data acquisition. We provide a view to bioimaging platforms for customised methodologies that employ machine learning techniques to analyse IAV liquid inclusions. In addition, we compare data obtained based on images acquired as one focal plane, a z-projection or as a volume, using antibodies in fixed cells or tagged viral proteins as proxies for inclusions in order to quantify selected thermodynamic processes. This is critical, given the extensive studies assessing the formation of condensates to understand physiological and disease mechanisms and performing large screens to understand how they can be modulated [39,40,41,42]. 

Establishing a clear correspondence between imaging acquisition, processing, thermodynamic changes, and phase transitions in our system is crucial. We found that when observing significant phase transitions, the methods can be used interchangeably, while small changes considerably vary based on the methodology employed. These findings will inform decision making regarding whether to invest in resource-intensive 3D imaging acquisition and machine learning techniques, which demand substantial data storage and analysis, or whether a more time-efficient 2D imaging approach suffices without compromising the accuracy and robustness of the obtained data.

Lastly, we explore the impact of using antibodies or fluorescently tagged proteins on data, considering the perspective of using wild-type or attenuated viruses in our experiments, as we found the data obtained to be significantly distinct. We advocate that these basic analyses are the of paramount importance when studying biomolecular condensates in a wide range of research fields.

## 2. Results

In our previous study, we merged live cell imaging with thermodynamic measurements to uncover that it is possible to impose phase transitions in viral inclusions that reduce IAV replication [8]. Utilizing live cell imaging, we quantified the material properties of these condensates, revealing molecular movement freedom and structural order, as achieved by others [1,29,30,31,32,33,34,43]. Using Z-projections of fixed images, we analysed thermodynamic parameters, such as morphology, topology, nucleation density, Cdense, and Cdilute, along with Gibbs free energy of transfer (ΔG) [37,44,45]. ΔG is a key determinant of a process feasibility, spontaneity, and stability shifts [34,38,46], with a lower ΔG indicating the stabilization of the system. Combining both approaches has proved insightful, as we demonstrated that an increase in the strength of interactions within the components of IAV liquid condensates triggered a phase transition that resulted in their spontaneous hardening that had several readouts, including a ~−2000 J·mol^−1^ decrease in ΔG. The same was not observed by varying the temperature or concentration of components within IAV liquid inclusions [8]. These finding are crucial, suggesting that targeting IAV liquid inclusions could pave the way for new antiviral research avenues. 

This earlier study set the groundwork and provided data, obtained from the Z-projections of images acquired through the volume of the cell, to explore potential technical challenges in studying biomolecular condensates using microscopy-based image analysis. To answer this question, we still used IAV inclusions as a model, but we had to use an infectious virus in which viral ribonucleoproteins (vRNPs) were fluorescently tagged and formed liquid viral inclusions. For this study, we used a fluorescently tagged version of IAV strain A/Puerto Rico/8/1934 (A/PR/8/34, H1N1, henceforward PR8-WT), in which mNeongreen was fused to segment 3, which encodes the subunit PA of the trimeric viral RNA-dependent RNA polymerase (RdRp, consisting of PB2, PB1, and PA; Figure 2A) giving rise to a PA-mNeongreen virus (henceforward PA-mNG). The PA-mNG virus is attenuated, which results in a roughly 2log difference in viral production when compared to the wild-type counterpart (Figure 2C). When compared to PR8-WT, it shows a delayed progression of infection (Figure 2B) and reduced levels of viral proteins (Figure 2D). vRNPs reach the cytosol later and viral inclusions form later in infection, being modestly visible at 7 hpi and well visible only at 10 hpi, rather than at 8 hpi (Figure 2B) as in the case of PR8-WT. After their emergence, these inclusions behave similarly to PR8-WT, fusing, dividing (Figure 2E), displaying liquid properties, reacting to temperature (Appendix A), and increasing with concentration (Figure 2B). 

In addition, we determined that liquid inclusions formed upon infection with the PA-mNG virus may be hardened, such as those formed in WT-PR8 infection [8]. As stated above, for the WT-PR8 virus, we subjected infected cells to several treatments, including changing temperature, concentration, and the number and strength of interactions, and found that the latter led to hardened IAV inclusions that no longer exhibited liquid behaviour (i.e., did not fuse or divide, lost the ability to undergo internal rearrangements, and failed to respond to shocks). To increase the strength and number of interactions within IAV inclusions, we used the drug nucleozin [8]. Nucleozin is a compound that oligomerises all forms of NP [47,48,49], having affinity for three different sites in NP [49]. As a result, it chemically polymerises NP, either free or in vRNPs, in a reversible manner [47]. This mechanism acts to inhibit IAV replication in cultured cells and in a mouse model of influenza infection [50] and was previously described as a novel class of influenza antivirals targeting the viral protein NP. IAV can evolve escape mutant viruses by mutating a tyrosine to a histidine in position 289 of NP (NP-Y289H) [49]. Therefore, in our previous work, we used nucleozin as a tool to enable us to establish the proof of principle that it is possible to harden IAV liquid inclusions by increasing the number and type of intra- and inter-vRNP interactions through the polymerisation of NP [8]. For the PA-mNG virus, hardened IAV inclusions are detected by the loss of spherical shape due to the increased aggregation of such inclusions over time, by incubation with nucleozin (Figure 2F), as well as by the lack of ability to fuse and divide (Figure 2G). 

Hence, although being an attenuated virus, the use of the PA-mNG PR8 virus is appropriate for the purpose of this study as it aims to compare the introduction of artefacts during imaging acquisition and data analyses. Indeed, the PA-mNG PR8 virus offers a unique opportunity to study different variables in methodological imaging analyses, such as whether it is different to analyse a 2D single focal plane, Z-projection, or a 3D reconstituted cell. In addition, it also allows for the comparison of fixed versus live imaged cells and the assessment of how fluorescent proteins versus proteins stained with antibodies (in this case, to mark vRNPs) impact results. 

### 2.1. Analyses of Biophysical Traits from Images of One Focal Plane with Different Z-Positioning

The first question we asked was whether imaging one focal plane from the bottom, middle, or the top of the cell gave rise to distinct results when mapping the intracellular biophysical traits of viral condensates (Figure 3A,I). For this analysis, we used the inherent fluorescence of mNeonGreen and studied liquid inclusions in two conditions: cells infected with PA-mNG PR8 virus for 10 h in complete media plus 2 h in DMSO (Figure 3A–H), which produces IAV liquid inclusions of normal size, or treated with nucleozin for 2 h (Figure 3I–Q), which give rise to larger structures as a result of aggregated inclusions. As seen in Video S1, treatment of infected cells with DMSO does not affect the liquid properties of IAV inclusions that are able to fuse and divide and relax to a spherical shape. Conversely, nucleozin treatment renders IAV inclusions more rigid, as they are able to stick together but not to coalesce into a sphere. Measurements of the ratio of the concentration of vRNPs in the cytosol and the nucleus (Ccytoplasm/Cnucleus) was higher in the bottom than the middle or top of cells (0.64 ± 0.2 relative to 0.48 ± 0.2 or 0.40 ± 0.1, mean ± standard deviation (SD) for bottom, middle, or top, respectively; Appendix A), without altering the area (Figure 3B, Appendix A) or the aspect ratio (Figure 3C, Appendix A) of the inclusions. The concentration of vRNPs in the inclusions was significantly different, with the top inclusions containing a lower concentration of vRNPs, as measured by Cdense (20.75 ± 7; 22.07 ± 8 and 11.49 ± 8 AU, mean ± SD for bottom, middle or top., respectively) (Figure 3D, Appendix A), despite not showing differences in Cdilute (Figure E). In addition, there are differences in free energy (ΔG = −RT*In*K, in which K $=\frac{Cdense}{Cdilute}$), with the top being consistently less thermodynamically stabilised or in energetically less favourable condition, as shown by an increase in ΔG when compared to the bottom (Figure 3F, ΔG, mean ± SD, for the top; −2004.8 ± 1149 J·mol^−1^ relative to −3465.4 ± 996 J·mol^−1^ in the middle). Furthermore, the number of inclusions was significantly increased at the bottom (151.6 ± 63, mean ± SD), relative to the middle (96.8 ± 58, mean ± SD) (Figure 3G), but not to the top. The alterations are, however, very modest. 

Treating cells with nucleozin provided the expected changes published in [8]. Inclusions become larger structures as a result of oligomerisation (Figure 3I), with an increase in Cdense (Figure 3K), and the stabilisation of the system (Figure 3M), which resulted in fewer inclusions (Figure 3N). Interestingly, in terms of aspect ratio (Figure 3J), using one focal plane does not significantly lead to irregularly shaped inclusions, but there is a consistent increase in the standard deviation, which is higher upon nucleozin treatment at the top (Figure 3J). Also, there was no reduction in Cdilute, as previously observed [8], upon nucleozin treatment (Figure 3L). All the data can be compared in Appendix A.

When comparing the analyses of different focal planes within the same cell, data on inclusions were consistent regardless of analysing images from the bottom, middle, or top, especially for nucleozin-treated samples. Exceptions are observed for Cdense, which presents the lowest values for the bottom and for the number of inclusions, in which more inclusions are detected at the bottom comparatively to the top but not to the middle. 

Altogether, the data show that although significant changes were detected, especially when analysing smaller inclusions, the differences are modest. Interestingly, despite having differences in the number of inclusions, the nucleation density, (ρ = $\frac{numberofinclusion}{CytoplasmArea}$, µm^−2^), which is the number of inclusion per area, did not significantly change, being 1.17 ± 0.4; 1.04 ± 0.4; or 0.90 ± 0.4 µm^−2^ considering the bottom, middle, or top).

### 2.2. Analyses of Biophysical Traits Using Z Project

In our subsequent investigation, we sought to compare the outcomes of using a single focal plane against compressing all images in the Z stack into a Z-projected image using sum of slices as the type of Z-projection. For the direct comparison with one focal plane images, our results show that at 10 hpi, the area (0.29 ± 0.04 µm^2^, mean ± SD), aspect ratio (1.47 ± 0.05, mean ± SD), and ΔΔG (−2326.4 ± 507.4 J·mol^−1^, mean ± SD) are within the same range of values obtained for one focal plane, despite the Cdense and Cdilute providing 2log magnitude difference (Figure 4A–F, Appendix A). The Cdense and Cdilute are calculated as arbitrary units that directly correlate with the mean fluorescence intensity that is brighter in the case of a Z-projected image. Mean fluorescence intensity was used as a proxy of the concentration of cell, nucleus, cytoplasm, and cytoplasmic inclusion [37,51,52]. Data differing significantly relate to the number of inclusions, which is more abundant in z-projected images (216 ± 94 inclusions per cell, in opposition to a maximum of 151.6 ± 63 for the bottom of the single focal plane), and this is not surprising (illustrated in Figure 1 and Figure 2). All these comparisons can be found in Appendix A. When nucleozin was added for 2 h, the data obtained were also consistent with data from one focal plane (Figure 4G–L). There was an increase in the area (1.35 ± 0.5 versus a maximum of 0.72 ± 0.3 µm^2^ µm^2^ for the top focal plane), an increase in the Cdense (from 446 AU in the Z-projections, and for one focal plane, a range between 28 AU on the bottom, to 42 AU on the top), and a mild increase in the aspect ratio. All data showed that nucleozin stabilised the system although to different degrees (2371.8 J·mol^−1^ in Z-projections, and a range of 1694.4 at the bottom, to 3358.2 at the top J·mol^−1^ for one focal plane).

The findings suggest that when examining single focal planes or Z-projections in situations that lead to significant shifts in behaviour, both methods yield similar trends. Therefore, these approaches can be used interchangeably in such cases. However, for minor changes, our results indicate that the choice of analysis methodology can affect the interpretation and data outcome. For example, to analyse subtle differences occurring in the material properties of viral inclusions, as those that may happen during the course of infection (either the maturation or loss of fluidity), it may be necessary to use 3D reconstitutions for measuring biophysical traits, as using both single focal planes or Z-stacks provide contradictory results. In addition, the results should be complemented using measurements of viscosity, surface tension, viscoelasticity, and macromolecular diffusion, as well as mechanical properties that require live-cell imaging approaches. 

### 2.3. Biophysical Traits Obtained Using Z Projected Images Differ from Infections with PA-mNG PR8 Virus and PR8-WT

We next sought to compare the biophysical data of the Z-projections with those obtained for PR8-WT that we published recently [8]. To achieve this, we conducted two separate sets of experiments. The first set involved varying the concentration of vRNPs over a time course of infection, as previously achieved [8] (Figure 2B). The second set focused on analysing several time points after the addition of the drug nucleozin, up to a duration of 2 h (Figure 2F). These diverse conditions enabled us to examine the effects in situations involving both small and larger condensates. For the increase in the concentration of vRNPs, we capitalised on the increase in vRNP levels in the cytosol during infection [53]. We also show a measure of the increase in the cytosolic concentration of vRNPs, providing the ratio of Ccytosolic/Cnucleus in Appendix A, in which the vRNP concentration in the cytosol relative to the nucleus increases up to 16 hpi. For infection with the PA-mNG virus, viral inclusions increase with size (Figure 4A, Appendix A), becoming more irregularly shaped (decreasing circularity and roundness) as infection progresses (Figure 4B, Appendix A, Appendix A). We explained that this change in shape was due to viral inclusions that were shown to interact with microtubules and the endoplasmic reticulum, conferring movement, and to inclusions being able to undergo fusion and fission events. We also observed a progressive steady increase in Cdense from the time liquid inclusions appear at 7 hpi (603.09 ± 236.0 AU, mean ± SD) until 16 hpi, which is the time at which the value of Cdense reaches a maximum level (Figure 4C, Appendix A). The same is also observed for Cdilute (Figure 4D, Appendix A). This indicates that the critical concentration (the concentration at which vRNPs reach a critical maximum point) occurs around 16 hpi. The system did not undergo changes in stability, as observed by the lack of variation of free energy (Figure 4E, Appendix A). Interestingly, the number of inclusions did not significantly change during the course of infection (Figure 4F, Appendix A). These results are consistent with the increase in cytosolic vRNP, which led to bigger sized inclusions that increase in Cdense until 16 hpi, becoming modestly destabilised in terms of the free energy of the system, which increases mildly. 

From the time of nucleozin was added, viral inclusions increased in size until 30 min after adding nucleozin (0.46 ± 0.2 until 1.34 ± 0.51 µm^2^, mean ± SD; Figure 4G, Appendix A) and became slightly more irregularly shaped (1.37 ± 0.04 to 1.55 ± 0.07 AU, mean ± SD; Appendix A). The value of Cdense also increased until 30 min post nucleozin addition, and ΔΔG reduced dramatically until that time point (from −1972.3 ± 252 to −4543.4 ± 621 J·mol^−1^, mean ± SD) without observed changes in Cdilute (Figure 4I–K, Appendix A). As expected, the number of inclusions reduced dramatically between these time points (from 298.4 ± 168 to 50.2 ± 43; Figure 4L, Appendix A). These results are consistent with the hardening phenotype of the nucleozin treatment, which stabilises the system, leading to larger aggregated inclusions. The thresholds were reached 30 min after adding nucleozin.

We then proceeded to compare the results obtained for the PA-mNG PR8 virus with those published for PR8-WT [8]. In the case of PR8, as the progeny vRNP pool enters the cytosol, the viral inclusions showed an increase in size (from 0.172 ± 0.04 to 0.289 ± 0.06 µm^2^, mean ± SD, Figure 4M, Appendix A) while maintaining a similar aspect ratio (Figure 4N, Appendix A), leading us to hypothesise that there is movement of the inclusions along microtubules, as stated above. To validate this assumption, we compared the aspect ratio of viral inclusions in the absence and presence of nocodazole (which abrogates microtubule-based movement). The data in Appendix A show that in the presence of nocodazole, the aspect ratio decreases from 1.42 ± 0.36 to 1.26 ± 0.17, supporting our assumption. The concentration of vRNPs inside the condensates (Cdense) increases until 8 h post-infection (hpi) (Figure 4O, Appendix A), accompanied by a rise in the diluted cytosolic phase (Figure 4P, Appendix A). Both parameters stabilise thereafter, indicating that the critical concentration is reached around 8 hpi. Notably, the free energy reaches its lowest point at 6 hpi (−1799.0 ± 623 J·mol^−1^, mean ± SD) and experiences a mild destabilisation (increase in ΔG) thereafter (−1139.8 ± 382, −1131.2 ± 444, and −833.8 ± 342 J·mol^−1^ @ 8, 12, and 16 hpi, respectively, mean ± SD) (Figure 4Q, Appendix A). The number of inclusions remains constant (Figure 4R, Appendix A). These findings are consistent with the notion that the increase in cytosolic vRNP leads to larger-sized inclusions while overall maintaining a similar concentration until 8 hpi, time at which the system reaches a plateau.

Next, we tracked how nucleozin affected IAV liquid inclusions in PR8-WT infections for different periods ranging from 5 min to 2 h relative to the data for PA-mNG. We observed similar results as in all other conditions above under nucleozin treatment. Nucleozin-treated inclusions became larger (from 0.284 ± 0.04 without nucleozin to 1.02 ± 0.18 µm^2^, mean ± SD, with 2 h treatment; Figure 4M, Appendix A) and formed a multi-shaped meshwork, which was reflected in the increased aspect ratio calculations (Figure 4N, Appendix A) Interestingly, Cdense increased dramatically (from 2125.8 ± 0.09 AU, mean ± SD, without nucleozin to 3650.0 ± 0.03 AU, mean ± SD, with 2 h nucleozin), Figure 4U) and Cdilute decreased. The system became more stable after 20 min treatment (from 728.1 ± 213 AU, mean ± SD without nucleozin to 398.6 ± 94 AU, mean ± SD after 2 h treatment, Figure 4V), suggesting that the system, rather than producing liquid inclusions, has formed a classical complex that is more rigid. Importantly, these structures were energetically more stable, with lower free energy (from −1711.1 ± 397 J·mol^−1^ without nucleozin to −5388.4 ± 808 J·mol^−1^ 2 h post nucleozin addition, mean ± SD; Figure 4W, Appendix A). As in all conditions, the number of inclusions decreased with the treatment (from 310.5 ± 133 to 38.1 ± 34 after 2 h treatment; Figure 4X, Appendix A). These results are consistent with the hardening phenotype of nucleozin, which stabilises the system, leading to bigger aggregated inclusions. The thresholds were reached 20 min after adding nucleozin.

Overall, although the system behaves similarly whether using PA-mNG virus or PR8-WT virus, and there are critical differences relative to the timing of threshold (16 hpi for PA-mNG and 8 hpi for PR8-WT). The alterations in Cdilute upon nucleozin treatment and the differences in aspect ratio are greater when using nucleozin in the case of PR8-WT. These differences point to two hypotheses: PA-mNG viral inclusions were assessed using the inherent fluorescence of vRNPs (marked in PA) rather than antibody staining like the ones in PR8-WT (highlighting NP), or to the fast response of the system relative to the increased concentration of vRNPs in the cytosol of cells infected with the PR8-WT virus. 

### 2.4. Using Antibodies versus Using Fluorescently Labelled Material Changes Readouts

Given the differences referred to above obtained from the calculation of the biophysical parameters upon analysing incremental concentrations of vRNPs and adding nucleozin for the two viruses PA-mNG PR8 and PR8-WT, we questioned whether using an antibody to highlight vRNPs might generate different results from vRNPs tagged using a fluorophore. For this, the PA-mNG PR8 virus is an ideal candidate, as we may use its inherent fluorescence, or an antibody, to detect the mNeonGreen that is fused to the PA (as a proxy of vRNPs). Furthermore, we can expand the observations by using an antibody against NP (another proxy for vRNPs), which may bind to several molecules of NP on vRNPs to test whether antibody staining against PA or NP provides distinct results (Figure 5A). As a note, each vRNA has a similar structure, but they vary in length from 890 to 2341 nucleotides. For each vRNP, the different RNA sequences are encapsidated by molecules of NP every 21–24 nucleotides along their length, and one unit of the RdRp bound to the base-paired RNA termini. The 3’ and 5’ termini are partially complementary and hybridise, forming a panhandle structure [54]. This structural variation implies that vRNPs can have approximately 35–95 molecules of NP and 1 of PA, depending on their size. As a result, using NP or PA as a proxy for vRNPs may yield significantly different sizes and normalised fluorescent intensities, which is a crucial aspect to consider when the aim is to analyse viral inclusions, which are sites where vRNPs accumulate.

To present the changes that may result from using a tagged virus either opposed to or in concert with fluorescent antibodies in 2D planes, it is paramount to understand how the analysis of a complete 3D cellular volume affects the thermodynamic parameters of IAV inclusions. We therefore introduced a modification to our analytical pipeline and included the volume of the cell using Imaris analysis software, version 10.0.0 (Videos S3 and S4, vRNPs labelled with PA fluorescence). This volumetric analysis estimates the fluorescence in 3D space of each inclusion, rather than compiling Z-stacks to collapse three-dimensional inclusions to 2D estimations. Our results show that using the inherent fluorescence of the mNeonGreen added to PA or antibodies to label either mNeonGreen or NP does not impact in quantifications of area, sphericity (aspect ratio), Cdense, or Cdilute (Figure 5B–E, Appendix A). Interestingly, detecting vRNPs with antibodies resulted in a modest reduction in the stability of the system, exclusively upon nucleozin treatment (−1994.01 ± 460.14 J·mol^−1^ for fluorescent PA in opposition to −1532.17 ± 790.26 and −1638.89 ± 495.75 J·mol^−1^ for the two antibodies; Figure 5F, Appendix A). The data show that these different methodologies may be used interchangeably for conditions originating more extreme changes in stability. However, mild changes may be best identified using fluorescently labelled vRNPs.

## 3. Discussion 

No model or analytical system is perfect. However, both model and analytical systems should be tailored to specific questions. In this manuscript, we comprehensively assessed differences in biophysical traits obtained using different imaging analysis modalities. For our case, we used IAV liquid inclusions, which are hypothesised to be sites of IAV genome assembly. In particular, we explored the impact of quantifying thermodynamic parameters using images acquired as one focal plane (Figure 3), Z-projections (Figure 4), and volumetric analyses of IAV inclusions stained by antibodies or live tagged proteins (Figure 5). These thermodynamic parameters are very useful for knowing how a system is behaving under normal or perturbed conditions. 

Opting for a single focal plane proved advantageous due to its efficiency in image acquisition, processing, and reduced computational demands. However, as illustrated in Figure 3 and Figure 4, imaging various Z-planes yielded differing results. While it is possible to standardise which plane to acquire, this may overlook crucial information about the cellular environment of specific condensates, potentially introducing human error and bias. Acquiring only a single focal plane may also be challenged by the inherent variability in cell physiology, leading to inconsistencies in results. 

On the other hand, obtaining Z-projections or full volumetric views of the cell necessitates capturing Z-stacks, resulting in larger file sizes and considerably longer data acquisition and processing times. Our results indicate that Z-projected images do not confer additional advantages over single focal plane images. In the context of extensive screenings, initiating with single focal plane images could present a remarkable temporal advantage for our specific system and research question, contrary to prevalent beliefs in the field. These findings are particularly pronounced when coupled with live-cell imaging analyses.

However, discrepancies observed between the attenuated PA-mNG PR8 and the PR8-WT virus in the analysis of Figure 4, alongside the lack of a significant difference in results using a fluorophore or antibody to mark viral inclusions (Figure 5), suggest that concentration influences the temporal response to specific stimuli and conditions, catalysing the reaction time. It is well known that concentration plays a critical role in phase separation [55], including in the maturation of liquid inclusions and resulting in increased viscosity as a result of Ostwald ripening [56]. In this article, we speculate that it may be important in the reaction time for hardening, especially the utilization of compounds such as nucleozin that “stick” vRNPs together, increasing the strength and number of interactions. For nucleozin to have an effect, more vRNPs need to establish interactions via nucleozin, which is easier in a more crowded environment. At the moment, this hypothesis has not been experimentally validated. If validated, it suggests that usage of attenuated viruses should be evaluated, and that the differences in concentration were compensated either by increasing the time of infection or by investing in the development of fluorescent viruses with less attenuated phenotypes.

On the side of larger data files that require more processing power and data storage capacity, volumetric fixed or live cell imaging will offer more confidence in aspects, such as shape, aspect ratio, Cdense, and Cdilute. This type of analyses has become a critical tool for studying condensates in general. In our case, fixed images will suffice; however, if using live imaging, one can follow dynamic processes and contextualise them in the cellular environment. For viral infections, this is particularly relevant, as it will be possible to track virion production and different stages of the viral lifecycle. It is important to carefully weigh the pros and cons and conduct preliminary studies, like the one presented in this manuscript, to determine the optimal strategy for approaching the image acquisition and analysis of biomolecular condensates, aligning with the scientific question at hand. Additionally, preliminary analysis contrasting 2D versus z-projection versus 3D volumes, tag versus antibody, as well as fixed versus live, holds critical importance in building confidence in the data concerning the study of biomolecular condensates. Such analyses could have implications in other cellular analytical studies as well.

In summary, detecting minor changes necessitates a robust analytical modality. However, if in pursuit of significant changes, our manuscript demonstrates that simpler processes of data acquisition and analysis may confer an advantage.

## 4. Materials and Methods

### 4.1. Cells

A549 cells (CCL-185, American Type Culture Collection, Manassas, VA, USA) and Madin-Darby Canine Kidney (MDCK) (a kind gift of Prof Paul Digard, Roslin Institute, Edinburgh, UK) were cultured in high-glucose (4.5 g/L) Dulbecco’s modified Eagle’s medium (DMEM) (21969035, Thermo Fisher Scientific, Waltham, MA, USA) supplemented with L-glutamine (200 mM), penicillin–streptomycin solution 100×, and 10% foetal bovine serum (FBS). 

### 4.2. Virus Preparation and Infection

The influenza A virus (strain A/Puerto Rico/8/1934 H1N1) (PR8-WT) virus with an mNeonGreen tag on segment 3 was rescued by reverse genetics using a previously published protocol (1), adapted from a pdual PA-GFP (Segment 3-GFP) PR8 construct to integrate mNeonGreen in place of GFP. MDCK cells were diluted to a concentration of 5 × 10^5^ cells/mL in high-glucose DMEM supplemented with L-glutamine, penicillin–streptomycin solution 100X, and 10% FBS (complete DMEM).

0.25 µg of plasmid was used per influenza virus segment in 200 µL of jetPRIME buffer. 5 µL of jetPRIME transfection reagent was added to the mixture, and the mixture was added to cells. After overnight incubation at 37 degrees Celsius, cells were treated with trypsin and resuspended in complete DMEM.

293T cells were prepared in a 6-well plate. These cells were dislodged by forcefully pipetting the infected MDCK cells onto the 293T cells in the plate. These cells were transferred to a T75 flask and were incubated for 4–5 h at 37 degrees Celsius. Once adhered, the cells were washed gently with serum-free medium (SFM) (complete DMEM without FBS) and 10 mL of SFM with 0.14% bovine serum albumin (BSA) and 1 µg/mL TPCK trypsin (1:1000) (LS003750; Worthington Biochemical Company, Lakewood, NJ, USA) was added. The cells were then incubated for four days. After four days, the supernatant was aspirated, and the cells were pelleted by centrifugation at 3000 rpm for five minutes. The PA-mNG PR8 virus was gathered from the supernatant and subsequently aliquoted into Eppendorf tubes and frozen at −80 degrees Celsius.

Regarding infection, A549 cells were infected with the PA-mNG PR8 virus (1.6 × 10^7^ PFU/mL) at an MOI of 3 or of 10, as indicated. The virus was diluted in serum-free medium and added to cells in 24 well plates containing sterile cover slips or live imaging chambers, for fixed and live imaging, respectively. Cells were infected for 10 h before fixation or treatment with dimethyl sulfoxide (DMSO) or 5 µM nucleozin (N2790; Sigma-Aldrich, Darmstadt, Germany) in DMSO (D8414; Sigma-Aldrich, Darmstadt, Germany). The PA-mNG PR8 virus infection cycle is attenuated as a result of its fluorescent tag. As such, 10 h of infection with PA-mNG PR8 approximately corresponds to 8 h of infection with the wildtype PR8 virus in terms of the development of infection and replication.

Virus infections with influenza A/Puerto Rico/34/8 were performed at a multiplicity of infection (MOI) of 3. After 45 min, cells were overlaid with DMEM containing either 0.14% bovine serum albumin (for plaque assays) or DMEM supplemented with 10% FBS, 1% penicillin–streptomycin solution and 2 mM L-glutamine (for immunofluorescence, Western blotting and RTqPCR). To calculate viral titres, supernatants collected from infected cells were subjected to a plaque assay on MDCK monolayers.

### 4.3. Antibodies

Commercial antibodies were used for both the primary and secondary antibody staining for immunofluorescence microscopy. With regard to the primary antibody mix, anti-mNeonGreen mouse monoclonal antibody (32F6; Proteintech Europe, Manchester, UK) was used at a 1:500 dilution in PBS. The secondary antibody mix contained donkey anti-Mouse IgG (H+L) highly cross-adsorbed secondary antibody, Alexa Fluor™ 568 (A10037; Thermo Fisher Scientific), chicken anti-rabbit IgG (H+L) cross-adsorbed secondary antibody, Alexa Fluor™ 647 (A21443; Thermo Fisher Scientific, Waltham, MA, USA), and Hoechst 33342, trihydrochloride, trihydrate—10 mg/mL solution in water (H3570; Thermo Fisher Scientific, Waltham, MA, USA), all diluted 1:1000 in PBS.

In the primary antibody mix, we also used a homemade anti-nucleoprotein rabbit polyclonal antibody, used in a 1:2000 dilution in PBS. This antibody was received as a generous gift from Dr. Paul Digard, Roslin Institute, Edinburgh, UK. 

### 4.4. Immunofluorescence Microscopy

Cells were cultured on cover slips or live imaging plates. Conditions for fixed samples were attained by administering 4% paraformaldehyde to cells for ten minutes, followed by washing twice with phosphate-buffered saline (PBS). The cells were then permeabilised using 0.2% Triton-x100 (X100; Sigma-Aldrich, Darmstadt, Germany) in PBS for 7 min on a rocker. The cells were then washed twice in a blocking solution of PBS with 1% FBS. 

30 µL droplets of the primary antibody mix was added to parafilm, and cover slips were placed face down on the droplets for one hour at room temperature (RT) in dark conditions. The cover slips were returned face up to their respective wells and washed three times in blocking solution. The secondary antibodies were added for 45 min at RT, in the dark on a rocker. Cover slips were washed with blocking solution three times and with PBS twice before cover slips were mounted to microscope slides using Dako Faramount Aqueous Mounting Medium (S3025; Agilent Technologies, Madrid, Spain). The images of 2D and 3D PA-mNG infected cells alone and treated with nucleozin were acquired using a super-resolution spinning disk confocal (SoRa) microscope (Marianas SDC with SoRa, 3i, Denver, CO, USA) equipped with a 63× objective lens. The images in which PA-mNG, the anti-mNG antibody, and the anti-NP antibody were tested together were acquired using a spinning disk confocal microscope (Marianas SDC, 3i, Denver, CO, USA) equipped with a 100× objective lens.

### 4.5. Image Analysis

Images were processed using either custom ImageJ ((Fiji is just) ImageJ version 2.14.0/1.54f) macros or custom Imaris (version 10.0.0) machine learning toolkits (labkit extension version 0.3.11) and then analysed using R analytical scripts (R studio version 4.1.0). In the case of ImageJ, the images were analysed and segmented using the sum of slices, to project the z-stack into 2D for quantification of areas. This ImageJ pipeline has been previously described by [8]. 3D volumetric analyses were carried out in Imaris using the labkit extension for segmentation. Nuclei and inclusions were identified and quantified using the surfaces tool in Imaris. Using the topology and thermodynamic analyses described by [8], the dilute phase of the cytoplasm was quantified by measuring the signal intensity of the cytoplasm with the nucleus and inclusion signals removed. Data were obtained through from two independent biological replicates in the case of all. 

### 4.6. Quantification and Statistical Analysis

Data were analysed using the R statistical package (R version 4.1.0), but the visualization of body weight changes was performed in GraphPad Prism 9.4.1 (681). To quantify thermodynamics and topological variables, we extracted imaging data using an Image J custom plugin or exported the data through Imaris® and a custom R analytics pipeline. For particle tracking, coarsening assay, photoactivation, photobleaching and shock treatments, we compared two groups: cells treated with DMSO and those treated with Ncz. After data transformation in R, we assessed for the homogeneity of variance. Homogenously distributed data were assessed by parametric test using either one-way ANOVA to analyse independent variables followed by a post hoc analysis by Tukey multiple comparisons of means, or a *t*-test for the comparison of two groups only. When the data were not homogenous, we used non-parametric analysis with statistical levels determined after Kruskal–Wallis Bonferroni treatment. For simplicity, the details of the test used for each experiment are included in the figure legends. In our case, when two groups were compared, they were not homogenously distributed, hence a non-parametric analysis was performed instead of a *t*-test. Alphabets above each boxplot represents the statistical differences between groups. Same alphabets indicate lack of significant difference between groups while different alphabets infer a statistically significant difference at α = 0.05.

## Figures and Tables

**Figure 1 ijms-24-15253-f001:**
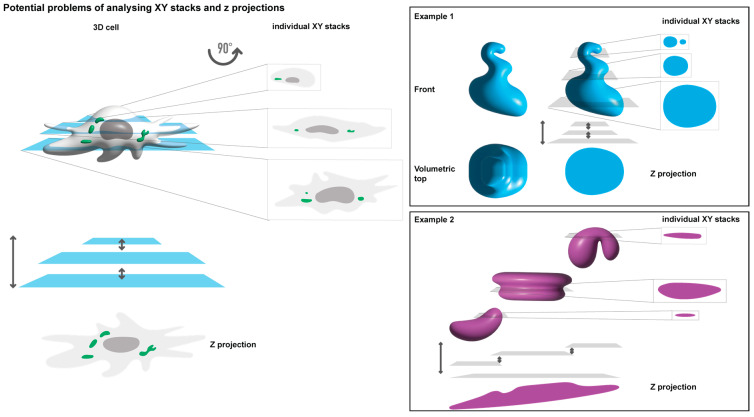
Image analysis in cells presents numerous challenges. Depicted image of a 3D cell containing biomolecular condensates (in green), the traditional acquisition of 2D images through the z-stacks throughout the entire volume of the cell (left). Below there is an illustration of how a Z-projected image of the individual stacks merging would look like (Z projection). Object quantification and shape may differ significantly based on whether they arise from a single focal plane. A Z-projection of has been 3D reconstituted. Examples of idealised objects (blue and magenta) are shown on the right.

**Figure 2 ijms-24-15253-f002:**
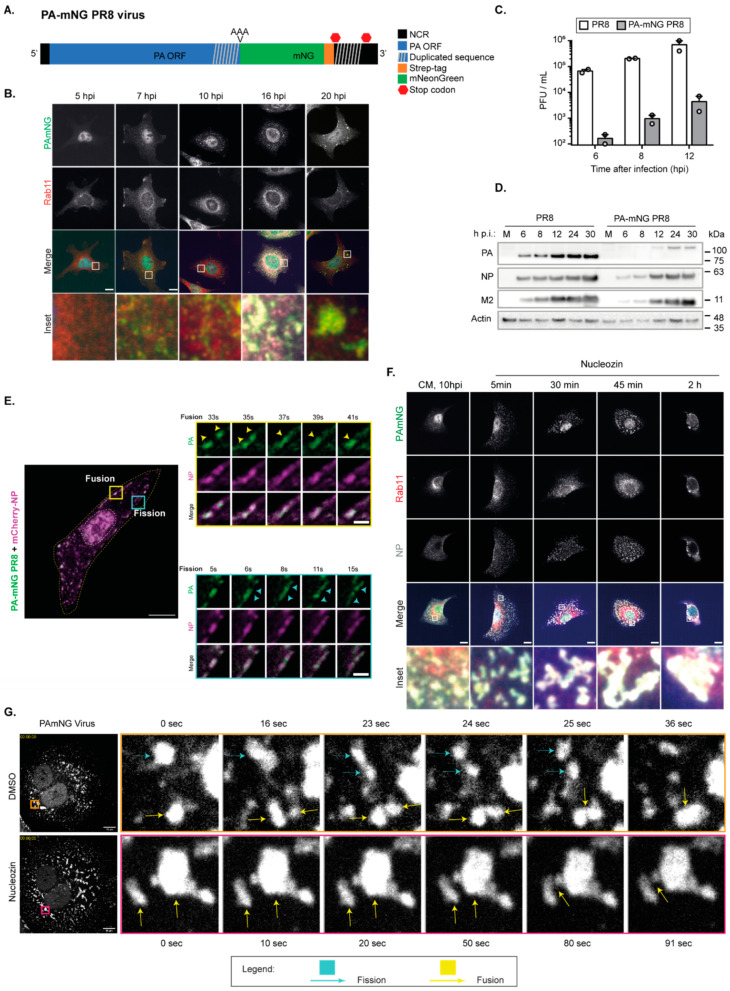
The PA-mNG PR8 virus is attenuated in comparison to the PR8-WT virus but produces viral inclusions with identical liquid properties. Of note, the liquid properties of PR8-WT have been described in [8,28] and will not be covered further in the present manuscript. (**A**) Representation of segment 3 from influenza PA-mNeonGreen PR8 virus to identify changes made to segment 3 for the insertion of the fluorescent tag (PA-mNG PR8; abbreviations: NCR: noncoding region; ORF: open reading frame; Strep: streptavidin). (**B**–**D**). A549 cells were infected at a multiplicity of infection (MOI) of 3 and at indicated timepoints. (**B**) Analysis by immunofluorescence using antibody staining against Rab11 and NP as a proxy for vRNPs. Scale bar = 10 µm. (**C**) The supernatants were used to determine viral production by plaque assay and plotted as plaque forming units (PFU) per millilitre (mL) ± standard error of the mean (SEM). Data were from two independent experiments. (**D**) The cells were used to determine the levels of the expression of the viral proteins PA, NP, and M2 and the cellular actin in cell lysates by Western blotting. (**E**) A549 cells were transfected with a plasmid-encoding mCherry-NP and co-infected with the PA-mNG PR8 virus for 16 h, at an MOI of 10, and were imaged under time-lapse conditions. Insets highlight vRNPs/viral inclusions in the cytoplasm in the individual frames. The dashed white and yellow lines mark the cell nucleus and the cell periphery, respectively. The yellow/cyan arrowheads indicate the fission/fusion events, respectively, and movement of vRNPs/ viral inclusions. Scale bar = 10 µm. Bar in insets = 2 µm. (**F**) A549 cells were infected at an MOI of 10 for 10 h, and after this time were treated with DMSO or 5 µM of nucleozin, a vRNP pharmacological modulator (stimulating NP polymerisation), and at indicated times fixed and analysed by immunofluorescence using antibody staining against Rab11 and NP using 63x magnification (scale bar 10 µm). (**G**) Infected cells were treated with DMSO or 5 µM nucleozin for 2 h, before live imaging for 2 min (Videos S1 and S2). Fusion (yellow arrow) and fission (cyan arrow) events are observed in infected cells treated with DMSO but not with nucleozin. Images taken at 63× magnification (scale bar 10 µm).

**Figure 3 ijms-24-15253-f003:**
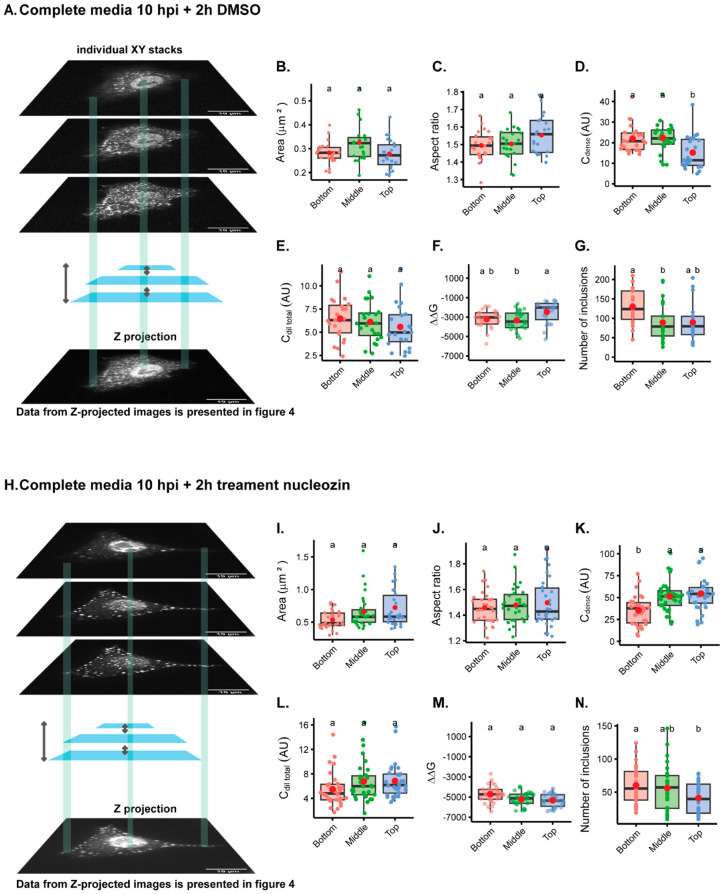
Analysing different single focal planes through Z reveals mild differences in biophysical data. A549 cells were infected at an MOI of 10 with the PA-mNG PR8 virus for 10 hrs and DMSO (**A**–**G**) or (**H**–**N**) 5 µM of nucleozin were added for 2 h before fixing. Cells were processed for immunofluorescence analysis, using the inherent mNeonGreen fluorescence (n = 23–30 cells). Each dot is the average value of a measured parameter per cell. Above each boxplot, same letters indicate no significant difference between them, while different letters indicate a statistical significance at α = 0.05 using one-way ANOVA, followed by Tukey multiple comparisons of means for parametric analysis, or Kruskal–Wallis Bonferroni treatment for non-parametric analysis. All the mean values calculated for the thermodynamics parameters have been included in Appendix A. Abbreviations: AU, arbitrary unit; CM, complete media; NCZ, nucleozin. (**B**,**I**) Boxplot of viral inclusion area (µm^2^) per cell. Statistical data by Kruskal–Wallis Bonferroni treatment. (**C**,**J**) Boxplot of aspect ratio of inclusions. Statistical data by one-way ANOVA followed by Tukey multiple comparisons of means. Aspect ratio = length of major axis of viral inclusions/length of minor axis of viral inclusions. (**D**,**K**) Boxplot of vRNP concentration within inclusions (C_dense_ (AU)). Statistical test statistical data by one-way ANOVA, followed by Tukey multiple comparisons of means. (**E**,**L**) Boxplot showing C_dilute_ (AU). Statistical data by one-way ANOVA followed by Tukey multiple comparisons of means. (**F**,**M**) Boxplot of ΔΔG (J·mol^−1^). Statistical data by one-way ANOVA followed by Tukey multiple comparisons of means. (**G**,**N**) Boxplot showing number of viral inclusions per cell. Statistical data by one-way ANOVA followed by Tukey multiple comparisons of means.

**Figure 4 ijms-24-15253-f004:**
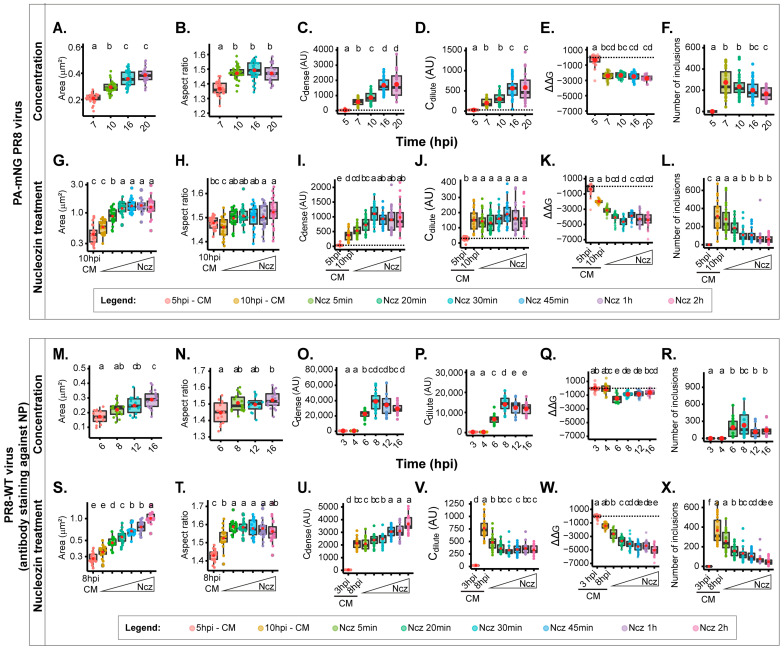
Biophysical traits obtained with Z-projected images vary in PA-mNG PR8- and PR8-WT-infected cells. Biophysical calculations in cells infected with the PA-mNG PR8 virus (**A**–**L**) or PR8-WT virus (**M**–**X**) upon altering the concentration (**A**–**F**,**M**–**R**) by taking advantage of the time course of infection as indicated or upon altering the type/strength of vRNP interactions by adding 5 µM of (Ncz) at 10 hpi during the indicated time periods (**G**–**L**,**S**–**X**). All data, area, aspect ratio, C_dense_, C_dilute_, free energy, and number of inclusions, were represented as boxplots. Above each boxplot, same letters indicate no significant difference between them, while different letters indicate a statistical significance at α = 0.05 using one-way ANOVA, followed by Tukey multiple comparisons of means for parametric analysis, or Kruskal–Wallis Bonferroni treatment for non-parametric analysis. The data of PR8-WT virus are only used here for comparative terms. (**A**,**G**,**M**,**S**). Boxplot of viral inclusion area (µm^2^) per cell; statistical data by Kruskal–Wallis Bonferroni treatment. (**B**,**H**,**N**,**T**). Boxplot of aspect ratio of inclusions. Statistical data by one-way ANOVA followed by Tukey multiple comparisons of means. (**C**,**I**,**O**,**U**). Boxplot of vRNP concentration within inclusions (C_dense_ (AU)). Statistical data by one-way ANOVA followed by Tukey multiple comparisons of means. (**D**,**J**,**P**,**V**). Boxplot showing C_dilute_ (AU). Statistical data by one-way ANOVA followed by Tukey multiple comparisons of means. (**E**,**K**,**Q**,**W**). Boxplot of ΔΔG (J·mol^−1^). Statistical data by one-way ANOVA followed by Tukey multiple comparisons of means. (**F**,**L**,**R**,**X**). Boxplot showing number of viral inclusions per cell. Statistical data by one-way ANOVA followed by Tukey multiple comparisons of means.

**Figure 5 ijms-24-15253-f005:**
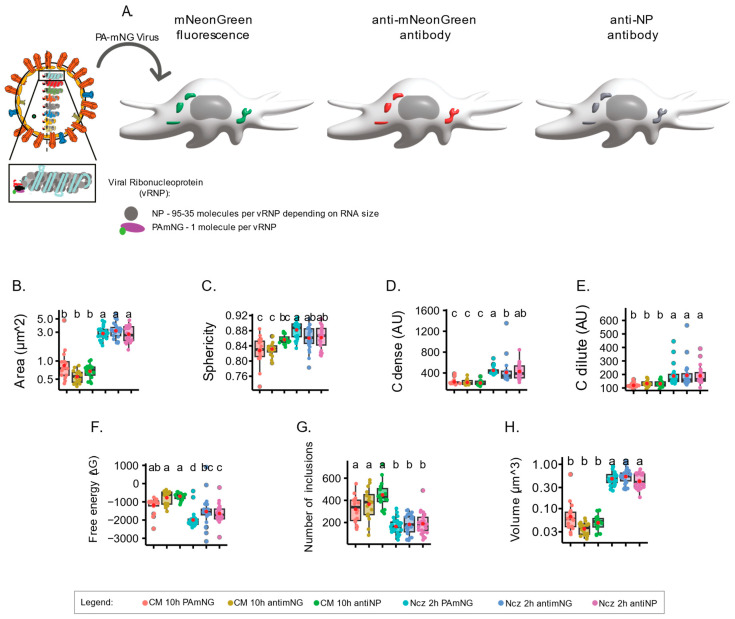
Analyses of biophysical parameters in 3D reconstituted cells where viral inclusions are stained with inherent fluorescence or antibody against mNG or NP provide data with similar biophysical traits. Biophysical calculations in cells infected with the PA-mNG PR8 virus at an MOI of 3 at 10 hpi and upon incubation with DMSO or 5µM of nucleozin (Ncz) for 2 h with viral inclusions highlighted by the fluorescence of mNeonGreen (mNG) or antibodies against mNG (to mark PA) or NP (antiNP). All data, area, aspect ratio, Cdense, Cdilute, free energy, number of inclusions, and volume, were represented as boxplots. Above each boxplot, same letters indicate no significant difference between them, while different letters indicate a statistical significance at α = 0.05 using one-way ANOVA followed by Tukey multiple comparisons of means for parametric analysis or Kruskal–Wallis Bonferroni treatment for non-parametric analysis. (**A**). Depicted image of the experimental design in which viral inclusions in cells infected with PA-mNG PR8 virus are detected with mNG fluorescence antibody to highlight PA or NP. The volume of the cell is analysed. (**B**). Boxplot of viral inclusion area (µm^2^) per cell; statistical data by Kruskal–Wallis Bonferroni treatment. (**C**). Boxplot of sphericity of inclusions. Statistical data by one-way ANOVA followed by Tukey multiple comparisons of means. (**D**). Boxplot of vRNP concentration within inclusions (Cdense (AU)). Statistical data by one-way ANOVA followed by Tukey multiple comparisons of means. (**E**). Boxplot showing Cdilute (AU). Statistical data by one-way ANOVA followed by Tukey multiple comparisons of means. (**F**). Boxplot of ΔG (J·mol^−1^). Statistical data by one-way ANOVA followed by Tukey multiple comparisons of means. (**G**). Boxplot showing number of viral inclusions per cell. Statistical data by one-way ANOVA, followed by Tukey multiple comparisons of means (**H**). Boxplot showing the volume (µm^3^) of viral inclusions per cell. Statistical data by Kruskal–Wallis Bonferroni treatment.

## Data Availability

All data, codes, and images will be available upon request to the corresponding author.

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
