# Peer review of "Challenges in Imaging Analyses of Biomolecular Condensates in Cells Infected with Influenza A Virus"

_ijms, 2023, doi:10.3390/ijms242015253_

Round 1

Reviewer 1 Report

In this manuscript, Temitope et al. discuss the analysis methods and technical limitations of microscopic imaging of droplet data. I agree with the limitations outlined in the paper. Currently, the field of phase separation research is thriving. An increasing number of researchers consider the formation, fusion, and separation of droplets as direct evidence of phase separation. As a systematic analysis method is crucial, this paper effectively examines the drawbacks and limitations of microscopic imaging. However, solely relying on improvements in analysis may not significantly advance research. The primary focus should be on employing higher-resolution precision microscopes, such as STED microscopy, to enhance resolution. Multiple approaches should be used to comprehensively investigate various aspects.

The analyzed aspects of the article are intriguing, but there are several questions that the authors need to address:

1. The authors' imaging is achieved through overexpression. Overexpression often leads to the rapid production of a large amount of protein, making it prone to aggregate formation. This state significantly differs from the physiological conditions within cells. Overexpression experiments might lack relevance in the context of phase separation research. What is the state of cells after being infected by the influenza A virus under physiological conditions? Can condensates be observed through imaging in this physiological context?

2. For endogenous proteins and those with low expression levels, such as transiently expressed proteins or those prone to rapid degradation, improving the resolution of microscopic imaging is crucial. Currently, there are user-friendly commercial software options available, such as Imaris. It would be valuable to compare the analysis using commercial software with the methods proposed by the authors to identify any differences.

3. How can the authors' methods be utilized to analyze the binding of two proteins that move rapidly?

Reviewer 2 Report

I found this manuscript very hard to follow and so I think that if the authors wish to get their message clearly over to an audience less familiar with the methods employed then the authors should give fuller explanations of precisely what is being measured and how the data are collected. I recognise the repeated claims that the manuscript is ‘technical’ but still it should be able to be read without too much difficulty. It is a pity in the way that the manuscript is presented tables can be displayed over more than a single page, and figure legends also can go over pages. These are not the clearest way to present the results. These problems might well have come from forming a PDF at the journal.

I am not competent on the assessment of the thermodynamic assessment the authors carry out: the delta G and delta delta G assessments derived from the equation delta G=- RTInK; nevertheless, I am concerned that ‘the system’ is not in equilibrium and so equilibrium-based thermodynamics may not be being applied in an appropriate fashion. It is not obvious that since lnK has been given as ln(Cdense/Cdilute), why taking logs is of any benefit.

The abstract promised ‘a forward looking perspective on using machine learning techniques’ but I do not see that being addressed in any detail here.

The introduction takes a very long time to get to the crux of the problem that the authors are addressing. I think that a much more concise introduction would be much better (and the authors might also consider being much more concise in the description of the results as well).

The use of a fluorescently tagged tagged virus ribonucleoprotein is a powerful tool in the experiments described. It is concerning (but not surprising) that the tagged virus PA-mNG PR8 is attenuated and is delayed in the production of virus polypeptides and the yield of virus. I presume the authors maintain that for the type of analysis they are carrying out that any attenuation is irrelevant since the paper is about imaging and not about the virus. Nevertheless, as I see in Figure 2, there have been no panels of wt-PR8 (that could be ‘pseudo’-tracked by Rab11) shown here. The authors can comment on why this is not needed. The title of the figure though would indicate that a comprehensive set of data for wt-PR8 should be given.

Figure 2 presents some effects of the application of nucleozin. Whilst nucleozin is described in the text, in the ligure legend it is described as ‘a vRNP pharmacological modulator’. The way nucleozin acts in a cell is probably worth describing more fully than as ‘a well-known sticking pharmaceutical compound’ (Line 238) – whatever that means. 

The description of different Z-positioning is described but, and here the authors use the term ‘stabilised’ (Line 287), but it is not clear what this means. Nor it is clear what the authors mean by ‘the dispersion in the number of inclusions’. Is this variability in the number of inclusions? The authors also ought to make it clear how the Cdense and Cdilute are, in principle, actually estimated from their images.

In this same section, it was not clear on the final paragraph of this section on different Z-planes it is stated that delta G is affected the most, but delta G is dependent on the ratios of Cdense and Cdilute and so this ratio between the two is the key perhaps, and not the natural log of the ratio.

The next section was analysis of the images projecting the Z-images onto a plane. The results are displayed in Fig 4. Here in the legend it is said that there is a change in concentration, but it is not stated what it was the concentration of that was being altered. The conclusion at the end of this section needs to be more thoroughly explained. The authors present that ‘for minor changes, our results indicate that the choice of analysis methodology can affect the interpretation and data outcome’. This needs a deeper analysis than is presented in the results to illustrate the conclusion.

The following section covers a comparison with published data and the results are shown in figure 4. In the text again, the concentration was not defined (line 366). Here it is stated that the inclusions become more irregularly shaped, but it is not clear how this was assessed since it is the aspect ratio that was presented, which does not imply irregularity. The authors observed that Cdense increased until it reached a threshold, but it is not clear what ‘threshold’ this might be. I wondered if threshold (L378) was the correct term for what they were wanting to write, perhaps the authors mean ‘a maximal level’ – a threshold being something to be crossed. Likewise, it is not clear what was meant by ‘the critical concentration’ (L379), nor on L384 what was meant by ‘becoming modestly destabilised’. 

Th authors introduce the concept of Csat and this requires a much fuller explanation and what the difference is between a stable complex and a liquid condensate, presumably the loss of a dynamic factor. One question that arises with the Cdilute reaching a stable low level is whether there is still RNP formation or NP or PA etc. synthesis when the effect of nucleozin has been observed. 

The next step was to use antibody to detect changes in the sub-cellular distribution/’condensation’ of the RNPs. The results are shown in Figure 5 and presented in Table 3. These results are not dealt with in a thorough fashion in the presentation of the results. A more detailed description is needed for the reader to be able to easily understand the authors reasoning. Here in Fig 5 the term ‘sphericity’ is used and not ‘aspect ratio’. It is not clear why this has been changed. The arrangement of the order of the data perplexing – in order to easily compare the different ways by which the RNPs are detected these should be grouped on the graphs, rather than ordering by minus and plus nucleozin.  Here the authors state that they are doing volumetric analysis (discussion lines 528 to 529) but this needs to be explained much more fully.

In the discussion valid points are raised but on lines that concentration plays a critical factor in the temporal response to stimuli and these catalyse reaction times. My feeling is that this catalysis was not established.  

Overall, I think that there is merit in the approach that is being taken but the authors ought to revise the manuscript considerably to enable to reader to gain most out of what is being presented. My feeling is that the extent of change needed is not simply changing a few words but a complete assessment of whether every paragraph and each figure legend is written in the most straightforward way, some are but many are not. In my opinion, the figure legends need a lot of attention.

Some minor points

Throughout: the word ‘data’ is sometimes used as singular and sometimes used as plural.

L179, the words ‘through put’ should be the single word ‘throughput’.

L185, describe what is meant ‘X-Y’ of 250nm.

L206-207, the meaning of the words ‘Z-projection of volumetric cell’ is not clear. This whole sentence could also be better punctuated.

L219, the name of PR8 is not given in the standard format: A/PR/8/34.

L234, the term ‘mechanistic mode’ is hardly elegant.

L242, what is meant by ‘independent IAV inclusions’?

L311, the word ‘changed’ should be ‘change’

L323, as written the abbreviation CM means complete medium with nucleozin, but I think it means complete medium. 

L 553, the authors use the term ‘heavier data’, the authors might reword this since data have no mass. 

L628 to 630, these needs re-punctuating.

Lines 663 to 664, mice are referred to, but nowhere else. I am not sure that this was an essential part of the materials and methods. 

The authors should give some virological parameters, such as the yield of virus when produced for inoculation of the cells. 

It is not the quality of the English that this the problem, it is the clarity that needs considerable improvement. 
